# My Health Diary, a School-Based Well-Being Program: A Randomized Controlled Study

**Emanuela Rabaglietti [1], Giorgia Molinengo [1], Antonella Roggero [1], Antonella Ermacora [2], Laura Marinaro [3] and Franca Beccaria [2],***

[1] Department of Psychology, University of Turin, 10124 Turin, Italy; emanuela.rabaglietti@unito.it (E.R.); giorgia.molinengo@unito.it (G.M.); antonella.roggero@unito.it (A.R.)
[2] Eclectica Institute for Research and Training, 10125 Turin, Italy; ermacora@eclectica.it
[3] Epidemiology Unit, Prevention Department, 12051 Alba, Italy; lmarinaro@aslcn2.it
* Correspondence: beccaria@eclectica.it

**Abstract:** Background: A positive transition to adulthood entails developing the individual and social skills needed to cope with critical situations. The "My Health Diary" program was designed as a school-based and teacher-led intervention focusing on the active engagement of 12- to 13-year-old pre-adolescents. The study analyzes the role of several primary variables (psychological well-being, psychosomatic symptoms, health status), secondary variables (health-risk behaviors, prosocial behavior, academic success, physical and verbal aggression), and mediator variables of emotional and social skills in terms of empathic and social self-efficacy, and satisfaction with school. Methods: Sixty schools were involved, divided into control groups ($N = 29$) and intervention groups ($N = 31$). The program was administered only to the intervention group. Of the 2306 students at the baseline, 2078 were still involved at post-intervention 6 months later. Results: The program was not found to have significant effects on the primary outcome variables and most of the secondary variables. For the mediators, however, the association was stronger for the girls in the intervention group, and there was a statistically significant difference in the empathic skills shown by girls, who reported higher levels than boys. Conclusions: The program was found to have encouraging effects on some mediators and in enhancing socio-relational and emotional skills among pre-adolescents.

**Keywords:** school-based program; well-being; socio-emotional skills; pre-adolescence; health-promotion

## 1. Introduction

In adolescence, as Havighurst argued in his Developmental Tasks Theory [1], later described by Manning [2] and documented by numerous authors in more recent studies [3–5], boys and girls are required to acquire social, cognitive, and emotional skills to adequately address developmental tasks.

Adolescence is a time of transition from childhood to adulthood [6] when boys and girls are suspended between past and future in an attempt to overcome the bridge that separates them. Clausen [7] described these crucial moments of change as turning points and life markers. During this phase, adolescents reflect on and re-evaluate past decisions and choices, redefine themselves according to new roles and expectations, and consider the changes that may take place in their lifestyles. These changes can then affect adolescents' re-organizational identity process and development paths in the life-span [8].

A positive transition into adulthood not only means avoiding risky behaviors. It also entails developing the individual and social skills needed to cope with critical situations and challenges which take place at the physiological level and are associated with growing up [9–11].

Consequently, enhancing adolescents' individual skills in normal life conditions is essential from the standpoint of universal prevention and promotion of psychological and social well-being. The scientific literature states that health promotion programs should start in pre-adolescence to reinforce adolescents' socio-emotional and life skills in addressing developmental tasks [12–15].

In this framework, the World Health Organization (WHO) recommends that school health promotion and prevention interventions be adopted to help children and adolescents develop life skills, i.e., abilities for adaptive and positive behavior that enable individuals to deal effectively with the demands and challenges of everyday life [16]. These skills are grouped into three broad categories: cognitive skills for analyzing and using information, personal skills for developing personal agency and managing oneself, and inter-personal skills for communicating and interacting effectively with others. As Botvin points out [17–19], life skills play an important role during adolescence in promoting psychosocial well-being and preventing health-risk behaviors such as drug and/or alcohol use, violence, aggression, and delinquency. In this connection, Botvin LST (Life Skills Training) [20] is an internationally recognized, evidence-based program designed to reduce adolescent risk behaviors. By providing young people with effective social skills and self-management skills, LST decreases the motivation to become involved in risk behaviors and the vulnerability to social influences that support health-risk behavior. LST provides foundational skills for successful youth development through Social Emotional Learning (SEL) competencies. At its core, SEL focuses on young peoples' fundamental needs for motivation, social connectedness, and self-regulation as prerequisites for learning [21]. In fact, SEL teaches children and adolescents to recognize and understand their emotions, feel empathy, make decisions, and build and maintain relationships. A recent meta-analysis found that SEL programs improve mental health, social skills, and academic achievement [22], while a follow-up study found that school-based SEL interventions continue to benefit students for months and even years to come [23].

Promoting adolescents' psychological and social well-being is thus a priority for public health intervention; in fact, researchers and policymakers [24,25] have recently focused much interest in these areas. School plays an important role in pre-adolescents' and adolescents' lives. As PISA data [26] indicate, the school's function is not only to facilitate the learning of notions and concepts but also to promote well-being through social relationships. As Langford et al. [27] argue, school is the most appropriate place to implement interventions for promoting adolescents' psychosocial well-being.

Several studies [14,28,29] emphasize the importance of projects for acquiring socio-relational and emotional skills. In the short term, such programs can reduce school difficulties and bring about a general improvement in school climate. In addition, they can produce positive long-term effects, helping students to become motivated to complete college and learn how to make safer decisions involving their sexual and mental health. These aspects, in turn, can promote a more general well-being in adolescents [30,31].

Similarly, the Schools for Health in Europe network (www.schools-for-health.eu/she-network) sees schools as a prime setting for improving young people's health and psychosocial well-being. In particular, socio-emotional factors are fundamental to achieving the health goals and educational objectives that the school sets for its students [32].

Pre-adolescence offers an important window of opportunity for health promotion and prevention programs for two important reasons. First, because the onset of the types of health-risk behavior typical of young people (i.e., the consumption of tobacco, alcohol, and psychoactive substances) occurs mostly in adolescence [33,34]. Second, pre-adolescence is a period marked by significant physical and psychological changes, when it is important to start adopting healthy lifestyles to encourage positive development and tackle poor habits before they set in [13,35].

However, almost all universal health promotion interventions address adolescents and are chiefly concerned with preventing individual risk behaviors [36–38]. As several studies have shown, the most promising programs are those that focus simultaneously on a variety of risk and protection factors. These factors can act as mediators of problem behaviors by increasing adolescents' resilience [37,39], promoting positive family relationships [40,41], enhancing social and emotional skills [22,42], and

improving the school experience [43,44]. It should also be borne in mind that teenage girls and boys may differ in how they deal with these factors, which are associated with their psychological adjustment and resilience [21,22].

To the best of the authors' knowledge, no interventions to promote psychological well-being in the 6th, 7th, and 8th grades are currently being implemented in Italy.

Furthermore, as pointed out by the Italian Network for Evidence-Based Prevention (NIEBP—see http://niebp.agenas.it/default.aspx), the interventions which have proven to be effective in preventing adolescents' risky behavior and promoting health and well-being were developed outside of Italy, making them difficult to adapt and transfer to the country's socio-cultural context.

The "My Health Diary" [Diario della salute] initiative is a school-based and teacher-led program for the active engagement of 12 to 13-year-old pre-adolescents developed based on Botvin's LST (Life Skills Training) [20] theoretical framework.

This study hypothesizes that the "My Health Diary" program increases subjective well-being among pre-adolescents by providing them with the social and emotional skills needed to fulfill their potential and deal with the developmental tasks of adolescence. The study analyzed the outcome of the updated version of the "My Health Diary" program on 12 to 13-year-olds based on the following: (i) primary variables, such as psychological well-being, psychosomatic symptoms, and health status; (ii) secondary variables, such as health-risk behaviors (cigarette smoking, alcohol use, unhealthy eating habits, and sedentary lifestyle), prosocial behavior, and physical and verbal aggression; (iii) mediator variables for emotional and social skills such as empathic and social self-efficacy, and satisfaction with school (relationship with teachers, relationship with classmates, interest in school, school self-esteem).

## 2. Experimental Section

### 2.1. The Program

The "My Health Diary" program aims to enhance the empathic and social skills of pre-adolescents and their level of satisfaction with school. This enhancement could increase their subjective psychological well-being and health status, which could be associated with increased pro-social behavior and academic success, a less sedentary lifestyle, and a reduction in physical and verbal aggression, cigarette smoking, alcohol use, and unhealthy eating habits.

The part of the program designed specifically for pre-adolescents consists of five standardized interactive units: My Emotions; Beyond Stereotypes; Becoming Men and Women; Managing my Emotions; Others' Emotions (Table 1). Each unit lasts between two to four hours and is conducted by the teachers. After the first version of the program was evaluated, two units were reviewed to focus more on the ability to manage emotions, and specific references to health-risk behaviors were deleted.

**Table 1.** Program units.

| Program Component | Activities | Materials | Duration (h) |
|---|---|---|---|
| Unit 1—My Emotions | Presentation, brainstorming, role-playing, recall of experiences associated with emotions, drawing, class discussion | Colored hats, marking pens, Post-it® notes, papers, posters | 3–4 |
| Unit 2—Beyond Stereotypes | Presentation, group work, game, class discussion | Photos, Identity cards, posters | 2 |
| Unit 3—Becoming Men & Women | Presentation, role-playing, group work, homework, class discussion | Newspapers or magazines, posters | 2–3 |
| Unit 4—Managing my Emotions | Presentation, group work, feedback, class discussion | Marking pens, Post-it® notes, my horoscope card | 2 |
| Unit 5—Others' Emotions | Presentation, group work, stimulus game, class discussion | Marking pens, Post-it notes, ambiguous image cards | 3–4 |

In addition, the program includes a narrative booklet for teens that tells the story of four same-age students dealing with common developmental tasks such as identity formation, relationships with peers, conflicts with parents, and physical changes at puberty, as well as a narrative booklet for parents that describes the experience of two families with teenage children and focuses on communication between parents and children and their relationship during adolescence [45,46].

*2.2. Participants*

The program employed a cascade training approach, in which 28 health professionals from two Italian regions (Piemonte and Veneto) participated in a two-day training course. They then trained middle school teachers in their area, selected the schools to enroll in the study, managed program implementation, administered study surveys, and organized courses for parents. A total of 75 teachers participated in the training courses, and 22 courses were organized for 315 parents. In this study, we focused on the program's effect on the subjective well-being of the direct target, the pupils.

Sixty schools took part in the study during the year under investigation. The schools were divided at random into control groups ($N = 29$) and intervention groups ($N = 31$). The program was administered only to the intervention group. A total of 130 classes participated, 67 in the control groups and 63 in the intervention groups. Of the 2306 students at the baseline (T1), 2299 were still involved at post-intervention (T2), and 2078 subjects filled out the questionnaire both before and after the program (Figure 1). They were balanced by gender (51% male) and aged between 11 and 15 years ($M = 12$ years $\pm$ 42), while 94% were of Italian nationality and 45% were of high socio-economic status (SES). A high SES indicator means that both parents have at least a high school diploma, whereas a low SES means that both parents finished either primary or middle school.

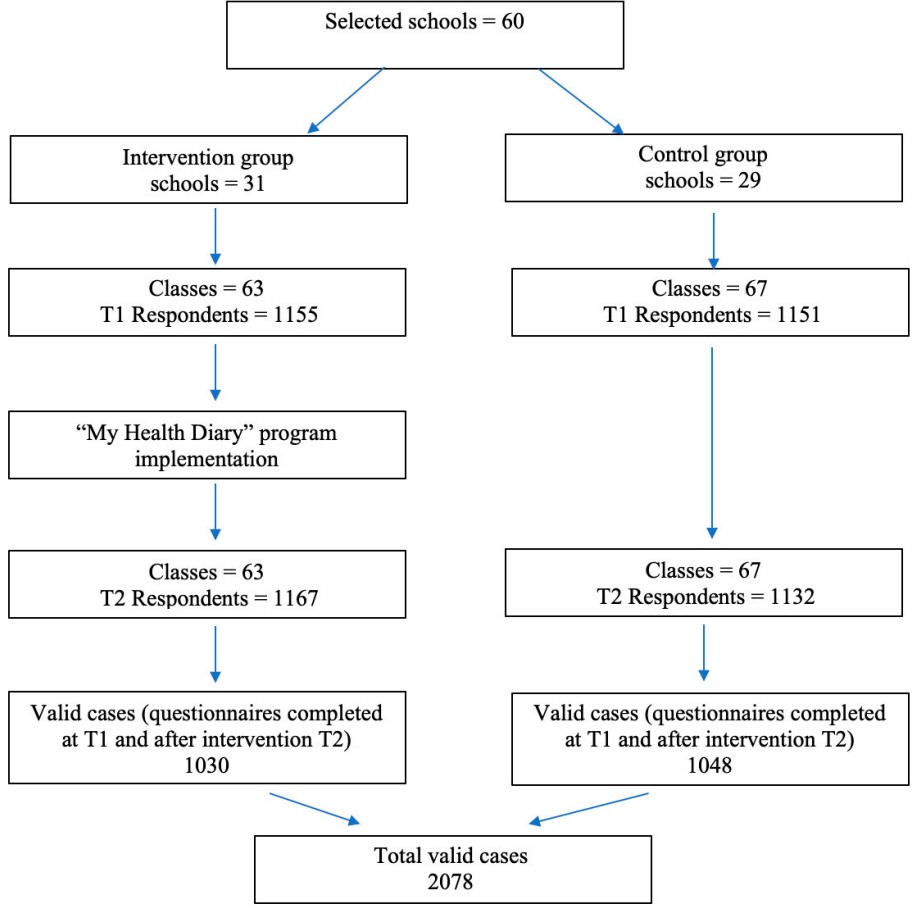

**Figure 1.** Study participants.

*2.3. Data Collection*

The study was approved by the Ethics Committee of the Santa Croce e Carle Hospital of Cuneo, Italy, and officially registered on the US National Institutes of Health clinical trials website (for more information, see https://clinicaltrials.gov/ct2/show/NCT02683811).

In accordance with Italian law and the ethical code of the Italian Psychological Association, students and parents were informed of the aims and methods of the study and provided written consent.

In the classroom, trained health professionals (THPs) informed students for whom consent had been obtained that their answers would remain anonymous. Students were also informed that their participation was voluntary and they could drop out at any time. The THPs administered the questionnaires in each classroom while teachers were not present. Students completed identical self-reported questionnaires before the intervention (T1) and 6 months after it ended (T2). The questionnaire took approximately 30 min to fill out.

All questionnaires were identified by a self-generated code used to link the pre- and post-intervention surveys [47].

*2.4. Measures*

2.4.1. Primary Variables

- *Subjective Psychological Well-being*: 4 dimensions (3 items for each dimension and a total of 12 items, *on* a 6-point Likert scale from 1—strongly disagree to 6—strongly agree: Autonomy, Cronbach's $\alpha$ T1 = 0.46; T2 = 0.41; e.g., *"I have confidence in my opinions, even if they are contrary to the general consensus"*; Environmental Mastery, Cronbach's $\alpha$ T1 = 0.40; T2 = 0.39; e.g., *"In general, I feel I am in charge of the situation in which I live"*; Positive relations with others, Cronbach's $\alpha$ T1 = 0.37; T2 = 0.44; e.g., *"People would describe me as a giving person, willing to share my time with others"*; Self-acceptance, Cronbach's $\alpha$ T1 = 0.39; T2 = 0.40; e.g., *"When I look at the story of my life, I am pleased with how things have turned out"*) from the Italian version of the Brief Psychological Well-Being Inventory [48–50]. This measure has been used for large-scale national surveys of adolescents [51,52] and was validated on an Italian population [53,54].
- *Subjective health status*: 1 item *"How would you rate your health?"* (on a 4-point Likert scale: 1—excellent, 2—good, 3—fairly good, 4—poor) from the Italian version of the HBSC (WHO/Europe-Health Behavior in School-aged Children, Copenhagen, Denmark) Questionnaire for 13-year-olds. This item has been used extensively for periodic HBSC surveys of 13-year-olds at international and national levels [55]. Self-rated health is a valid predictor of mortality and morbidity [56].
- *Psychosomatic symptoms*: 7 items (on a 5-point Likert scale: 1—rarely or never, 2—once or twice a month, 3—about every week, 4—more than once a week, and 5—about every day) from the Italian version of the HBSC Questionnaire for 13-year-olds—Symptom Checklist. These items ask students how often in the past 30 days they have suffered from headache/stomach ache/backache, irritability or bad temper, nervousness, sleeping difficulties, dizziness, and feeling low. The HBSC Symptom Checklist has been widely used for periodic HBSC surveys of 13-year-olds at international and national levels [55,57,58].

2.4.2. Secondary Variables

- *Risk behaviors*: cigarette smoking (2 items), alcohol use (2 items), unhealthy eating habits (2 items), physical inactivity in terms of frequency of sedentary behaviors (2 items) from the Italian version of the HBSC Questionnaire for 13-year-olds (on a 4-point Likert scale regarding frequencies of these behaviors in the past 30 days: 1—never, 2—1–3 days, 3—4–5 days, 4—every day). All these items have been commonly used for the HBSC international and national surveys of 13-year-olds for similar health promotion and prevention programs [59,60].

- *Prosocial Behaviors*: Prosocial Behavior Scale by Caprara et al. [61] (15 items, e.g., *"I try to help others"* on a 3-point Likert scale from 1—never to 3—often; Cronbach's α T1 = 0.67; T2 = 0.70). This scale has been widely used for national surveys [61,62], including some pre-adolescents [63–65]. It is also used to assess the level of adjustment/maladjustment in children aged 7 to 12 years with the indicators of social adaptability [62].
- *Physical and Verbal Aggression*: 20 items (e.g., *"I kick and hit or punch"; "I threaten others"* on a 3-point Likert scale from 1—never to 3—often; Cronbach's α T1 = 0.81; T2 = 0.82) from the Physical and Verbal Aggression Scale by Caprara et al. [61]. This scale has been widely used to survey pre-adolescents nationally and internationally [61,63–66].

### 2.4.3. Mediator Variables

- *Emotional and social skills*: Self-Efficacy in Regulating Positive Emotions (7 items, e.g., *"How well can you rejoice over your successes?"*—adolescent version; Cronbach's α T1 = 0.76; T2 = 0.80) and Negative Emotions (8 items, e.g., *"How well can you get over irritation quickly for wrongs you have experienced?"*—adolescent version; Cronbach's α T1 = 0.76; T2 = 0.79), Perceived Empathic Self-Efficacy (12 items, e.g., *"How well can you read your friend's needs?"*—adolescent version; Cronbach's α T1 = 0.86; T2 = 0.89), Perceived Social Self-Efficacy (13 items, e.g., *"How well can you live up to what others think or expect of you?"*—adolescent version; Cronbach's α T1 = 0.86; T2 = 0.88) from versions of Bandura's Self-Efficacy Scales [67] adapted and translated into Italian [68]. All items of the four self-efficacy domains are on a 5-point Likert scale: from 1—not well at all, to 5—very well. These scales have been used in large-scale national surveys of adolescents [69,70] and have good psychometric properties [68], which was confirmed by the reliability indexes in our study.
- *Satisfaction with school*: 4 scales (5 items for each dimension on a 4-point Likert scale from 1—never to 4—always: Relationship with peers, Cronbach's α T1 = 0.59; T2 = 0.57; e.g., *"I feel comfortable with my classmates"*; Relationship with teachers, Cronbach's α T1 = 0.57; T2 = 0.59; e.g., *"I can have a good dialog with my teachers"*; Interest in study, Cronbach's α T1 = 0.68; T2 = 0.71; e.g., *"I listen with interest to what is being explained to me"*; School Self-esteem, Cronbach's α T1 = 0.66; T2 = 0.63; e.g., *"I think I am a good student"*) taken from the School Situation Questionnaire (student version, QSS-S) by Santinello and Bertarelli [30]. These scales have been used in large-scale surveys of adolescents at a national level.

### 2.5. Statistical Analysis

A descriptive analysis was carried out on the variables. The chi-squared test was used to compare proportions and analysis of variance (ANOVA) to compare means. Specifically, repeated-measures ANOVA was used to evaluate pre- and post-intervention changes by gender and between the intervention and the control groups. Analyses were performed using IBM SPSS (version 25.0, Armonk, NY, USA).

## 3. Results

No effects of the intervention on primary outcome variables were found.

Results for subjective psychological well-being are shown in Table 2: both males and females reported a fair level of autonomy and there were no statistically significant differences, either relating to gender or to the group they belong to following the intervention.

**Table 2.** Effect of the intervention on subjective psychological well-being (repeated-measures ANOVA).

| | | Pre-Intervention (Means) | | Post-Intervention (Means) | | |
|---|---|---|---|---|---|---|
| | | Intervention Group | Control Group | Intervention Group | Control group | |
| Autonomy | M | 11.3 | 11.07 | 11.02 | 11.02 | Hotelling's T = 0.000; F (1980;1) = 0.73, *p* = 0.39 |
| | F | 10.9 | 10.7 | 10.7 | 10.8 | |
| Environmental mastery | M | 12.5 | 12.5 | 12.7 | 12.8 | Hotelling's T = 0.000; F (1956;1) = 0.86, *p* = 0.35 |
| | F | 12.4 | 12.4 | 12.6 | 12.4 | |
| Positive relationships with others | M | 12.7 | 12.6 | 12.6 | 12.7 | Hotelling's T = 0.000; F (1941;1) = 0.17, *p* = 0.68 |
| | F | 13.3 | 12.9 | 13.1 | 13.0 | |
| Self-acceptance | M | 12.1 | 12.1 | 11.9 | 12.1 | Hotelling's T = 0.000; F (1992;1) = 0.34, *p* = 0.56 |
| | F | 11.8 | 11.8 | 11.6 | 11.6 | |

Reported environmental mastery was also fair for both genders and control/intervention groups, and there were no statistically significant differences. In both groups, pre-adolescents reported positive relationships with others.

Lastly, boys and girls in both groups showed no statistically significant differences in self-acceptance. We consider this to be a positive result.

Most participants rated their health as good or excellent, and there were no statistically significant differences between the intervention and the control groups (*Chi-square* = 0.98, *df* = 3, *n.s*). As regards psychosomatic symptoms, there were no differences between the intervention and control groups in headache/stomach ache/backache (*Chi square* = 3.26, *df* = 2, *n.s.*), irritability/bad temper, nervousness (*Chi-square* = 4.76; *df* = 2, *n.s.*), sleeping difficulties (*Chi-square* = 0.84; *df* = 2, *ns*), and dizziness (*Chi-square* = 0.94, *df* = 2, *n.s.)*. The intervention and control groups differed significantly (*Chi-square* = 9.53, *df* = 2, *p = 0.009*) in feeling low (Table 3).

**Table 3.** During the last month (past 30 days), how many times have you experienced feeling low.

| | Pre-Intervention | | Post-Intervention | |
|---|---|---|---|---|
| | Intervention Group | Control Group | Intervention Group | Control Group |
| Never | 43% (430) | 43% (441) | 34% (348) | 34% (356) |
| 1–2 times a month | 29% (296) | 30% (308) | 31% (317) | 36% (379) |
| 1 or more times a week | 27% (276) | 28% (287) | 35% (358) | 29% (306) |

Regarding the secondary outcome variables, the intervention was found to have no statistically significant effects on satisfaction with school (*Chi-square* = 2.1, df = 3, *n.s.*), prosocial behavior (*Hotelling's T* = 0.001; F (1996;1) = 1.71, *n.s.*), physical and verbal aggression (*Hotelling's T* = 0.002; F (1934;1) = 3.54, *n.s.*), alcohol use (*Chi-square* = 1.001, df = 21, *n.s.*), alcohol abuse (OR = 1.023), physical inactivity and unhealthy eating habits. Though most of the adolescents in both groups did not smoke, there was a positive association over time between the control groups and smoking in the last 30 days (OR = 0.724; Table 4).

**Table 4.** During the past 30 days, have you smoked cigarettes?

| | Pre-Intervention | | Post-Intervention | |
|---|---|---|---|---|
| | Intervention Group | Control Group | Intervention Group | Control group |
| No | 98% (1003) | 97% (1013) | 96% (978) | 94% (975) |
| Yes | 2% (16) | 3% (28) | 4% (45) | 6% (62) |

Lastly, the intervention's effect on several possible mediators was evaluated (Table 5). These mediators included emotional and social skills in terms of self-efficacy in regulating positive and negative emotions, empathic self-efficacy, and social self-efficacy, as well as overall satisfaction with school as indicated by the relationship with peers and teachers, interest in study, and school self-esteem. The social self-efficacy of young people in both groups was quite high before and after the intervention. However, after the intervention, the association was stronger for the girls in the intervention group (*Hotelling's T* = 0.002; F (1873;1) = 4.48; *p* = 0.034). A statistically significant difference was also found for the same group of girls' empathic skills: after the intervention, they reported higher levels than boys (*Hotelling's T* = 0.002; F (1939;1) = 3.91; *p* = 0.048).

**Table 5.** Effect of the intervention on the mediators (repeated-measures ANOVA).

| | | Pre-Intervention (Means) | | Post-Intervention (Means) | | |
|---|---|---|---|---|---|---|
| | | Intervention Group | Control Group | Intervention Group | Control Group | |
| Self-efficacy, negative emotions | M | 25.3 | 25.4 | 25.8 | 26.1 | Hotelling's T = 0.000 F (1962;1) = 0.82, *p* = 0.09 |
| | F | 23.6 | 23.2 | 23.4 | 23.7 | |
| Self-efficacy, positive emotions | M | 28.3 | 28.3 | 28.4 | 28.6 | Hotelling's T = 0.001 F (2014;1) = 2.3, *p* = 0.13 |
| | F | 29.3 | 29.2 | 29.4 | 29.1 | |
| Empathic self-efficacy | M | 40.7 | 40.8 | 41.0 | 42.1 | Hotelling's T = 0.002 F (1939;1) = 3.9, *p* = 0.048 * |
| | F | 42.5 | 42.1 | 43.0 | 42.3 | |
| Social self-efficacy | M | 49.8 | 50.6 | 49.9 | 50.9 | Hotelling's T = 0.002 F (1873;1) = 4.5, *p* = 0.03 * |
| | F | 47.6 | 47.6 | 48.4 | 47.3 | |
| School self-esteem | M | 13.7 | 13.9 | 13.8 | 14.0 | Hotelling's T = 0.000; F (1937;1) = 0.93, *p* = 0.34 |
| | F | 13.3 | 13.2 | 13.1 | 13.3 | |
| Interest in study | M | 13.1 | 12.8 | 12.6 | 12.5 | Hotelling's T = 0.000 F (1913;1) = 0.29, *p* = 0.58 |
| | F | 13.5 | 13.4 | 12.9 | 13.0 | |
| Relationship with peers | M | 15.6 | 15.6 | 15.3 | 15.6 | Hotelling's T = 0.002 F (1939;1) = 5.59, *p* = 0.02 ** |
| | F | 15.6 | 15.7 | 15.5 | 15.3 | |
| Relationship with teachers | M | 14.0 | 13.9 | 13.3 | 13.1 | Hotelling's T = 0.000 F (1908;1) = 0.53, *p* = 0.47 |
| | F | 14.4 | 14.3 | 13.6 | 13.6 | |

Footnote: */** significant value.

No further statistically significant differences were found for the other self-efficacy mediators. As regards self-efficacy in managing one's own negative emotions and expressing one's own positive emotions, boys and girls in both groups felt that they were on average effective.

Lastly, young people reported being on average able to deal with school demands, and having a fairly good relationship with teachers, but did not show much interest in studying. Although the relationship with their classmates was quite good, it decreased post-intervention among the boys in the intervention group and among the girls in the control group (*Hotelling's T* = 0.00; F (1939;1) = 5.59; *p* = 0.018).

## 4. Discussion

"My Health Diary" is a universal prevention program addressed to pre-adolescents who regularly attend school and who in general are not involved in health-risk behavior. Positive evaluation of one's well-being is an important aspect in the life of the individual, and in adolescence has been shown to be associated with less involvement in risky behaviors and less psychological distress [71–73]. The study showed the intervention had no effects on the primary variables and that pre-adolescents reported medium-high levels of subjective well-being; these findings were in line with the national data from the HBSC study on the population of 11–13 year-olds [35].Although the good level of health and

well-being reported by the pre-adolescents is reassuring and consistent with the nature of the sample, results also revealed an increase in the frequency of feeling low in the total sample. This finding could be associated with new and increased demands from the surrounding environment, which call for more complex coping skills and resources on the part of young people. In accordance with national and international findings from the HBSC study [35,74], moreover, adolescents reported an increase in the frequency of somatic and psychological symptoms. It may be that respondents who claim to suffer from certain psychosomatic symptoms did not necessarily have a negative perception of their health and well-being. However, the increase in the frequency of feeling low was higher in the intervention group. This could be an effect of greater self-knowledge and awareness of one's internal emotional states following the intervention. Though this is an unexpected result, it confirms the findings of the evaluation study of the previous version of the program [45] and can be seen as positive in the light of the scientific literature stating that the individual's ability to recognize aspects of the self is a protective factor in the re-organization of boys and girls' identity, which is one of the main developmental tasks in adolescence [1,75,76].

As the sample of pre-adolescents is non-clinical but made up of young individuals who are capable of coping with the typical developmental tasks of their age (i.e., going to school, socializing with groups of friends, engaging in sports), a positive relationship can be hypothesized between psychological well-being and self-awareness, as indicated by some scholars [77,78]. A more introspective and critical attitude begins to emerge in pre-adolescence [79]: the young people in the intervention group, girls in particular, seemed to have acquired a greater ability to recognize their emotions, as well as more interpersonal and empathic skills when relating to significant others. While boys assign more importance to "doing things" together with their friends, girls establish relationships based more on intimacy, reciprocity, emotional sharing, and self-disclosure [80–83]. These aspects are fundamental to a more general socio-emotional competence, in turn, associated with better psychological adjustment and resilience [21,22]. Furthermore, while self-efficacy was quite high in both groups before the intervention, it increased afterward, among girls in particular. The "My Health Diary" program seems to have acted positively on another factor—empathic and social self-efficacy—which strengthened the ability, especially among girls, to cope with obstacles, challenges, and developmental tasks typical of their age.

Confirming the findings of the evaluation study of the previous program version [45], the "My Health Diary" program had no effects on secondary outcome variables (health-risk behaviors, prosocial behavior, physical and verbal aggression). An explanation for this can be sought like the program's target population. This population consists of pre-adolescents in the first grades of middle school: an age group that still has little involvement in and exposure to risk behaviors typical of adolescence which can affect their health and psychosocial well-being. Some risk behaviors for 12 to 13-year-olds are not related to the reorganization of identity and relationships with peers, as this occurs at a later age [10]. Consequently, the short period elapsed between the two data collections did not make it possible to detect significant effects of the intervention.

In this connection, some studies indicate that it is advisable to intervene early with respect to risk behavior involvement, emphasizing that the best age for prevention is immediately before young people start experimenting with risks such as cigarette smoking or alcohol consumption [84–86]. For this reason, a long-term evaluation could show different and more positive results. It can also be assumed that implementing the program for an older population that is more exposed to experimenting with risk behaviors could produce different results even in a short period of time. Furthermore, the small increase in the level of involvement in cigarette smoking found over time in both groups (more marked in the control group) can be interpreted as a "natural" increase in the experimentation that characterizes entry into adolescence and is thus not necessarily alarming.

We know, in fact, that many health-risk behaviors have been wrongly evaluated in the past, leading to confusion between what constitutes normal development, in which these behaviors are only transitory, and pathological development, in which these behaviors become ingrained over time [10,87].

Indeed, only a small percentage of young people who engage in health-risk behavior will persist with it into adulthood [34,88–90].

The intervention was found to not affect other behaviors identified as secondary outcome variables, such as verbal and physical aggression and prosocial behaviors. Most pre-adolescents showed good relationships with their peers and within their own life contexts. This does not mean that there are no risk factors for which action should be taken, but that from the standpoint of universal prevention pre-adolescents appear to be well equipped with the individual abilities needed to face the typical demands and relationships of their age. Finally, even the absence of effects on some of the associated variables considered in the study can be interpreted as testifying to pre-adolescents' emotional and social capacity. In fact, the "My Health Diary" program was credited with helping pre-adolescents strengthen these useful skills. Through the students' classroom involvement and reflection, "My Health Diary" enhanced the core of competencies needed to recognize and manage emotions, set and achieve positive goals, appreciate others' viewpoints, establish and maintain positive relationships, make responsible decisions, and handle interpersonal situations constructively. In the short term, the intervention has mainly affected the socio-relational and emotional components, as other programs based on an SEL (Socio-Emotional Learning) approach [14,21,29] have done. Especially for girls, it promoted a better ability to manage social relationships both in general and with respect to empathic skills.

To sum up, and noting that the program's assumptions, operation, and objectives have been revised since the previous version [45,46], the results achieved, although not entirely satisfactory in terms of the effect of the intervention, are encouraging. It can be assumed, in fact, that the pre-adolescents involved in the project can face challenges and successfully overcome the developmental tasks typical of their age.

The study presented herein has several limitations. One is the brief follow-up time and the absence of a long-term follow-up data-collection. Changes in attitude and behavior take time and are the result of a complex interaction between multiple individual and environmental factors. Effects of health promotion interventions on pre-adolescents are not immediate, and thus cannot be observed with a short-term assessment. These effects could emerge at a later age, once boys and girls who participated in the study are faced with typical adolescent developmental tasks or are more involved in health-risk behaviors. In addition, the methodology used was exclusively quantitative. Given the complexity of the object of study, this could be a further limitation. Indeed, introducing qualitative methods in the research design [91] could have provided a broader perspective and made it possible to drill down into the intervention's effects on pre-adolescents.

## 5. Conclusions

School is an excellent setting for implementing programs promoting young people's development. It plays an educational role and is responsible for providing support to girls and boys so they are more capable of consciously addressing their life choices and coping adequately with the different developmental tasks they face while growing up.

The most promising interventions seem to be those that promote boys' and girls' life-skills and socio-emotional skills [14,17,23,29,92,93] and that use interactive methods to transmit content [87,94,95]. Overall, the "My Health Diary" program is innovative on the Italian scene as well as at the European level for this specific age group, given the effort made to evaluate the intervention and bring together a network of professionals (health workers, teachers, etc.) who work with pre-adolescents in the school setting.

The study's findings suggest several areas for future work. First, the "My Health Diary" program, created as a universal health promotion and prevention intervention for pre-adolescents from normative samples, could also be implemented as a selective prevention intervention for a population of pre-adolescents at psycho-social risk with fewer emotional and social skills. It would be equally important to extend the program to adolescents slightly older than the original target population, as during this period of life some attitudes and behaviors adopted by young people can expose them

to a greater risk of jeopardizing their health and their psychological and social well-being both in the short and long term [10,34,37,88].

**Author Contributions:** Conceptualization: F.B., A.E., L.M., E.R. and A.R.; Methodology: F.B., G.M., E.R. and A.R.; Formal Analysis, G.M. and E.R.; Investigation, F.B., A.E. and L.M.; Data Curation, G.M.; Writing—Original Draft Preparation, F.B., G.M. and E.R.; Writing—Review& Editing, F.B. and E.R.; Supervision, F.B.; Project Administration, F.B.; Funding Acquisition, L.M. All authors have read and agreed to the published version of the manuscript.

**Funding:** This project was funded by the Italian Ministry of Health, Center for Disease Control and Prevention (CCM) Program 2011 (Grant Number J19E11002100001) and the evaluation office of the Piedmont Region—Prevention Sector.

**Conflicts of Interest:** The authors declare no conflict of interest.

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
