# Peer review of "My Health Diary, a School-Based Well-Being Program: A Randomized Controlled Study"

_adolescents, doi:10.3390/adolescents1010003_

Round 1

Reviewer 1 Report

Overall, the topic is interested that its contents. The value added of the paper has been articulated for contribution to knowledge contribution. The practical implication of the paper is yet to be fully addressed. However, I found it rather challenging to unclear why school-based well-being, it is hard to reconcile the finding. The paper does not fully address the research hypothesis as well as some theoretical to fill gaps of research study.

1. The abstract does not clear about purpose of study, methods (data analysis) and conclusion (see 6-20, page 1).

2. The introduction does not provide a clear picture of the problem statement (line 44-100) objectives (line 100-124).

3. Literature. The authors should be added/provided more review section. This is leading to test the hypothesis.

4. Methods. Statistical analysis does not clear. This section should be more detailed how did you analyse technique (line 204-208).

5. Results. Table 5 should be more detailed about what value means, what values indicate. Moreover, should be tested the hypothesis results are required.

6. Discussion read well.

7. The conclusion should be more practical implications, limitation and future direction.

8. References should be followed journal guideline. Both text and the list references should be corrected. 

Reviewer 2 Report

Thank you for this paper entitled "My Health Diary, a school based well-being program: a randomized controlled study." This paper is impressive, and the size of the paper is acceptable. The introduction is adequate and widely discusses the different health model by fitting adolescents into it. However, the methods section lacks the necessary information. As the study is RCT, several pieces of information need to be added to the methods. My suggestions/ comments are as follows:

  1. What was different in the intervention group. Please add in the abstract.
  2. Page 2 (Line 77 and 96), please put a reference number instead of a web address.
  3. Please add if "My Health Diary" is based on any health model. On line 100, the paragraph starts with "According to this theoretical framework" but fails to state if this study is based on any health model discussed above.
  4. Please add the hypothesis of the study.
  5. Line 131 "60 schools took part in this edition of the program, divided according to a randomized procedure". Please explain this in detail. How were the schools separated into two groups? Also, explain how the study was blinded.
  6. Presenting only the results of the intervention is fine. However, you need to add what was experimented with. Please explain clearly "what made the intervention group different from the control group."
  7. Please include the response rate in both groups. Also, calculate the drop out rate.
  8. Because this study is an intervention study, please add a flowchart showing participants included in the study. You can use the CONSORT flow diagram to present the study participants.
  9. Line 233: Please erase the typo after unhealthy eating habits.
  10. Please make the conclusion shorter and try to refrain from using references.

Reviewer 3 Report

This is an interesting study with a big sample size. However, I also have a serious concern about possible ‘experimenter effects’ in the procedure.

Introduction:

“My Health diary” program should be described in more detail including a brief description of how teachers’ formation was developed, how variables included are improved during the intervention, and how the sessions are organized along with the ordinary curriculum in the school.

The program should be better described using a figure, including the blocks and variables worked in each block.

Description of the program could also be moved to the method section.

The introduction should end with clearly articulated expectations (hypotheses) that advance current knowledge and that drive the data analysis. Authors should also include the grounds for those hypothesis.

Method:

Please, clarify if the total sample included in the analyses is 2078.

The program is intended to work with 12 to 13 years old students. However, in the sample description the age of the participants is 11 and 15 years old. Did authors used a subsample of 12 to 13 years old? In that case, please explain or include a rationale for the inclusion of 14-15 years old participants.  

What was the exact number of sessions in the program?

Questionnaires:

The authors states that the required time to fill out the questionnaires was approximately of 20 minutes. It seems to be a limited time for the number of questionnaires used.

Instead of reporting general psychometric properties of the questionnaires in previous studies, authors should include psychometric properties in the current sample.

Please, include item examples in each measure.

Please, indicate how many time passed from the end of the intervention and the post- intervention measurement.

Did the control group received the formation after the study was completed?

My major concern comes as regards of the not description of the procedure: Who ‘carried out’ the pretest assessments? Who ‘applied’ the interventions)? The issue here is about possible experimenter effects and whether any blinding procedures were used. First, were the people (teachers?) giving the sessions, aware of the purpose of the study? Could this affect how they interacted with the students? How can this affect the results? Please, explain.

The second issue is potentially just as serious. How were the pre and post-test assessments carried out? In a group, or individually? And crucially, were the people giving out the assessments aware of the study hypotheses, and of whether they were assessing experimental or control group students? Medical research, and also previous developmental psychology research (for example, in the play tutoring field) have demonstrated the possible importance of these experimenter effects. They could be especially marked in individual assessments but could also operate in group assessments.

Especially when no precautions were taken, this could unfortunately invalidate the conclusions of the study, or at least, render them unreliable.

Results:

Authors have reported some gender differences in the results but no information about how gender may operates was given in the introduction and the importance to test for gender differences.

Discussion:

Given that, in comparison to the control group, the intervention group did not obtain better scores in the variables measures, more discussion is needed about potential problems of “My Healthy diary” program reaching its objectives.

 Conclusion:

Conclusion section is reiterative and includes mostly what the study was intended to analyze. This information is also include in the abstract and the introduction. I would advise to only answer the question: What is the take-home message of this paper? Here, some conclusive remarks may be specified by the authors for the general readership

Round 2

Reviewer 1 Report

Read well. All comments are revised and adequate.

Reviewer 2 Report

The authors have addressed all suggestions for their manuscript. Hence, I agree with publishing this research.

Reviewer 3 Report

I would like to thank authors for the revisions made. The programa is now much better described and conclusions are directly drawn in their results.